# Ethical considerations in the prehospital treatment of out-of-hospital cardiac arrest: A multi-centre, qualitative study

**Louise Milling**[1,2]*, **Dorthe Susanne Nielsen**[3,4], **Jeannett Kjær**[1,2], **Lars Grassmé Binderup**[5], **Caroline Schaffalitzky de Muckadell**[5], **Helle Collatz Christensen**[6], **Erika Frischknecht Christensen**[7,8], **Annmarie Touborg Lassen**[9], **Søren Mikkelsen**[1,2]

**1** Department of Anaesthesiology and Intensive Care, Prehospital Research Unit, Odense University Hospital, Odense, Denmark, **2** Department of Regional Health Research, University of Southern Denmark, Odense, Denmark, **3** Department of Geriatric Medicine, Odense University Hospital, Odense, Denmark, **4** Department of Clinical Research, University of Southern Denmark, Odense, Denmark, **5** Department for the Study of Culture, Philosophy, University of Southern Denmark, Odense, Denmark, **6** The Danish Clinical Quality Program, National Clinical Registries, Copenhagen, Denmark, **7** Centre for Prehospital and Emergency Research, Aalborg University Hospital and Aalborg University, Aalborg, Denmark, **8** Emergency Medical Services, Region North Denmark, Aalborg, Denmark, **9** Emergency Medicine Research Unit, Odense University Hospital, Odense, Denmark

* louisemilling@dadlnet.dk

## Abstract

### Background

Prehospital emergency physicians have to navigate complex decision-making in out-of-hospital cardiac arrest (OHCA) treatment that includes ethical considerations. This study explores Danish prehospital physicians' experiences of ethical issues influencing their decision-making during OHCA.

### Methods

We conducted a multisite ethnographic study. Through convenience sampling, we included 17 individual interviews with prehospital physicians and performed 22 structured observations on the actions of the prehospital personnel during OHCAs. We collected data during more than 800 observation hours in the Danish prehospital setting between December 2019 and April 2022. Data were analysed with thematic analysis.

### Results

All physicians experienced ethical considerations that influenced their decision-making in a complex interrelated process. We identified three overarching themes in the ethical considerations: *Expectations* towards patient prognosis and expectations from relatives, bystanders, and colleagues involved in the cardiac arrest; the *values and beliefs* of the physician and values and beliefs of others involved in the cardiac arrest treatment; and *dilemmas* encountered in decision-making such as conflicting values.

**Data Availability Statement:** Data cannot be shared publicly because of GDPR and the Danish Data Protection Act. Data access can however be applied for, for specific purposes e.g. researchers

need to verify the project results. Data access or transfer will be possible in accordance with GDPR and the Danish Data Protection Act after approval from the authorities. Approvals to transfer and grant data access must be applied through the Research Support Unit at Odense University Hospital at open@rsyd.dk.

**Funding:** LM has received funding from Trygfonden (https://www.tryghed.dk/om-os/about-tryghedsgruppen) and the Laerdal Foundation (https://laerdalfoundation.org/). The funders had no role in study design, data collection and analysis, decision to publish, or preparation of the manuscript.

**Competing interests:** The authors have declared that no competing interests exist

## Conclusion

This extensive qualitative study provides an in-depth look at aspects of ethical considerations in decision-making in prehospital resuscitation and found aspects of ethical decision-making that could be harmful to both physicians and patients, such as difficulties in handling advance directives and potential unequal outcomes of the decision-making. The results call for multifaceted interventions on a wider societal level with a focus on advance care planning, education of patients and relatives, and interventions towards prehospital clinicians for a better understanding and awareness of ethical aspects of decision-making.

## Introduction

Decision-making in out-of-hospital cardiac arrest (OHCA) treatment can be classified as a dynamic process involving a sequence of choices made in an environment that can change exogenously [1]. Several factors need to be considered by the prehospital clinicians in the decision to initiate, continue, terminate, or withhold resuscitation. The prehospital work environment, the presence of bystanders, and high levels of stress among others may influence the decision-making in resuscitation [2]. The prehospital clinician must incorporate ethical decision-making in medical and clinical decision-making [3]. When ethical principles collide with each other, with the law, or with medical factors, this creates challenges or even dilemmas. These challenges may not only lead to individual variations in medical treatment but also an increased emotional burden and moral distress for clinicians [4]. Ethical challenges and considerations in decision-making are described in complex in-hospital contexts [5]. The in-hospital setting often allows for conversations with peers, the patient, and the relatives, provides for knowledge of the patient's medical history, and perhaps the patient's wishes regarding treatment and resuscitation [6]. By comparison, out-of-hospital resuscitation is characterized by time pressure and a lack of peer consultation and background information. In both settings, the treatment of cardiac arrest is characterized by irreversibility in decision-making: If the treatment is terminated, the patient will most likely die.

Ethical considerations are a universal part of medical decision-making and have been described in both somatic, psychiatric, and health community settings [7–10]. The four bioethical principles proposed by Beauchamp and Childress are acknowledged as important parts of the treatment of OHCA in the European Resuscitation Guidelines [3]. They "1) Respect for autonomy (respecting the decision-making capacities of autonomous persons), Non-maleficence (avoid causing harm), 3) Beneficence (provide benefits and to balance benefits against risks) and 4) Justice (obligations of fairness in the distribution of benefits and risks)" [11]. These four principles [12] are universal, can be applied in all healthcare settings [13] and should guide decision-making for healthcare professionals [3].

The present study explored prehospital physicians' real-life experiences and ethical reflections in decision-making during OHCA. We aimed to elucidate the ethical considerations made in prehospital resuscitation.

## Methods

### Design and theoretical framework

We conducted a multisite ethnographic study on prehospital physicians' decision-making in OHCA focusing on ethical considerations using a hermeneutic-phenomenological design.

This approach focuses on the lived experience and is useful in studying sensitive subjects such as ethical dilemmas and end-of-life care [14]. Participant observations during resuscitation were combined with in-depth interviews with the prehospital physicians in charge during each OHCA. We used method triangulation with semi-structured interviews, observations and field notes. Method triangulation can be beneficial in confirming data and increasing the validity of the findings [15]. The study is reported according to the COnsolidated criteria for REporting Qualitative research (COREQ) guidelines [16] (See S1 File).

## Study setting

Only a physician can declare a person dead in Denmark. Paramedics and EMTs can, however, withhold or terminate resuscitation after consulting with a physician, but do not require consultation with a physician if the patient has a do-not-attempt-resuscitation order (DNACPR) or if the patient has verbally refused resuscitation [17].

The EMS system in Denmark consists of EMTs, paramedics, and prehospital physicians. The prehospital physicians are anaesthesiologists with prehospital training who are dispatched alongside an ambulance or requested as rendezvous by the EMTs or paramedics on-scene. The Mobile Emergency Care Unit (MECU) is manned by a prehospital anaesthesiologist assisted by a paramedic. The MECU, in which the researcher conducted her participant observations, is a large vehicle allowing an observer to be present and communicate with the anaesthesiologist.

## Data collection and sample

Data collection took place between December 2019 and April 2022. The observations were conducted by author LM on six different MECU bases for a total of 825 hours. These MECU bases were located throughout Denmark with approximately 100–200 km between each base, and 400 km between the two bases furthest apart. Denmark is a small country and the longest distance across the country is 500 km. Due to COVID-19 restrictions, data collection was interrupted between July 2020-December 2021. Participants were included by convenience sampling of the physicians who were on duty whenever author LM was present as an observer. All participants had the opportunity to opt out of the interviews, but none did.

We observed 22 prehospital cardiac arrest treatments in total, and subsequently interviewed the 17 physicians who were responsible for the treatment in each case. See Table 1 for participant characteristics. The median age was 55 years (IQR 44.5–62). The median interview duration was 37 minutes (IQR 30.5–47).

Table 1. Characteristics of interviewed participants (n = 17).

| Health region (n) | A | 3 |
|---|---|---|
| | B | 4 |
| | C | 2 |
| | D | 8 |
| Gender (n) | Women | 4 |
| | Men | 13 |
| Prehospital work experience years (median (IQR)) | | 11 (7–14.5) |
| Primary subspecialty | Intensive care | 5 |
| | Anaesthesia | 6 |
| | Mixed | 6 |

## Participant observations and field notes

The observation guide was developed using methods described by James Spradley [18] and consisted of themes such as a description of the people at the scene, verbal interactions, and physical interactions (See Appendix 3 in S3 File). Unstructured observations and informal interviews were collected during the entire observation period using field notes (See Appendix 3 in S3 File). During participant observations, author LM balanced between moderate and active participation alternating between the insider and outsider perspectives [18]. Spradley describes the moderate participant as a "watcher" from the outside never really gaining the same skill or status as the observed, whereas the active participant seeks to do what other people are doing to gain acceptance and learn the social rules [18]. During observations, the author LM at times became a part of the daily life of the prehospital physicians and the team of EMTs and paramedics. This meant engaging in tasks such as the daily check of the MECU and help carrying bags or equipment shifting the participation from moderate to active [18]. Likewise, the insider and outsider perspectives are described as natural roles an observer will fall into during participant observations [18]. The insider will experience situations in an immediate subjective manner, while outsiders will observe the situation from the outside. Participant observers will usually shift between the two perspectives continuously and at times be able to attain both perspectives [18]. This was also the case in this study. Author LM has a background as a physician and is experienced in the prehospital setting. She was employed as a PhD student at the time of data collection and has experience and training in qualitative data collection. Participants were informed of LM's employment as a PhD-student and the project being part of this PhD.

## In-depth interviews

The interviews were conducted using an individual in-depth approach [19] allowing the participants to describe their experiences and reconstruct their perception of an event freely [19]. LM interviewed the participants one-on-one as soon as possible after the OHCA treatment during the physician's shift at the ambulance station. Three interviews were conducted the day after the OHCA due to time constraints during the shift. The interview guide was designed to explore decision-making and ethical challenges (See Appendix 2 in S2 File). We encouraged the participants to discuss their feelings towards decision-making in OHCA during the interview. The interview guide consisted of questions exploring inductively the feelings and experiences of the participants combined with deductively derived questions based on existing literature on the subject of ethics in OHCA decision-making. The interview guide was tested on two physicians who were not part of the study before initiating interviews. The audio-recorded interviews were conducted in Danish to allow for detailed answers and were transcribed by a secretary ad verbatim.

## Data analysis

A thematic analysis was performed inspired by Braun and Clarke [20]. Thematic analysis has been widely applied in health research and is useful when analysing individual experiences, opinions, and views [21]. By reading and re-reading the transcripts, we became familiar with the data. A hybrid approach of theory-informed and data-driven deductive and inductive coding was used for analysing the interview data. We first performed an inductive line-by-line coding using NVivo (QSR International, Burlington, Massachusetts, USA) to highlight the meanings in the text. The codes and statements relevant to the ethical aspects of decision-making were grouped into preliminary themes inspired by the theory of bioethics in resuscitation as a theoretical framework [22]. These themes were reviewed again with grouping and

regrouping of the codes. Lastly, we defined and named the final themes. In an iterative process, we moved back and forth between the different steps during the analysis.

## Ethics

The study was assessed by The Regional Committees on Health Research Ethics for Southern Denmark (no. 20232000–23) and exempted from ethical approval. According to Danish Law, studies not conducted on humans or without the inclusion of biological material do not need ethical approval. All participants received written information about the purpose and method of the study before the observations. The participants were informed verbally and in writing about the study and about the possibility of withdrawing consent at any given time on the day of observation. All participants signed a written informed consent form before participating in the study.

## Trustworthiness

This study followed the principles of trustworthiness as described by Korstjent et al. [23]. Prolonged engagement and persistent observations were achieved by participating in long shifts. This heightened the credibility of the study. Additionally, both data-, investigator-, and method triangulation boosted credibility [23]. A field diary was kept during observations and data analysis, which heightened reflexivity [23]. Confirmability and dependability were achieved through an audit trail [23], where decisions made during data collection and data analysis were documented and discussed with a qualitative expert (author DN).

## Results

Overall, the ethical considerations described as part of the decision-making could be divided into three themes: Expectations towards and from relatives, bystanders, and prehospital and in-hospital colleagues involved in the cardiac arrest, the values and beliefs of the physicians and other involved parties, and dilemmas encountered in decision-making (Fig 1). See Table 2. The ethical considerations were experienced as an interrelated process where the prehospital physicians were urged to deliberate and weigh out considerations for all parties involved (Fig 2). Quotes supporting the themes are presented in Table 3.

### Expectations towards and from parties involved in the cardiac arrest

The physicians' reports contained a wide variety of expectations directly or indirectly influencing the decision-making. Data included expectations towards both patient prognosis and patient quality of life, but also the expectations expressed by other parties involved in the cardiac arrest treatment.

**Expectations about patient prognosis.** The physicians assessed the patient's prognosis by reflecting on the cardiac arrest setting and they emphasized that they deduced the patient's functional capacity and rehabilitation potential in part from the setting in which the patient was found. In the patients' homes, the physicians noted the presence of assistive devices, home oxygen concentrators, and care beds. In cases where such assistive devices were found, the physicians often concluded that the patient had a low functional capacity and a low rehabilitation potential, and the physicians were thus more reluctant to continue resuscitation. Likewise, the physicians mentioned the state of the patient's home as a proxy for the level of functional capacity i.e., what was perceived as an unkempt home indicated a low potential for rehabilitation. (Table 3, quote no. 1.1. and 1.2.)

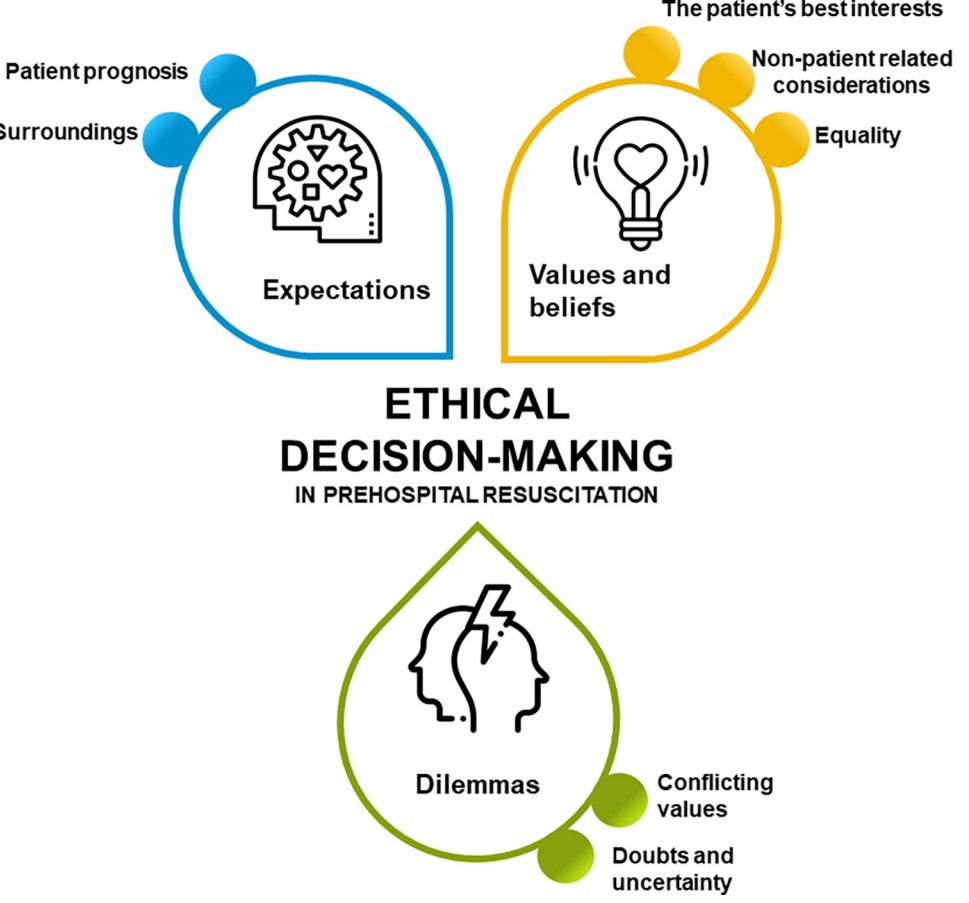

**Fig 1. Themes and subthemes.** Figure displaying the identified themes and subthemes.

The physicians described how patient care in locations such as nursing homes and sheltered accommodations influenced their expectations towards a bad prognosis, while cardiac arrest in public places was considered an indicator of a higher functional capacity and thus a better prognosis. Several physicians described how the patient's age influenced expectations towards comorbidity, functional capacity, and rehabilitation potential while stressing that age was just one part of many factors used to assess the patient. On the outer extremes of the age spectrum, i.e., very young and very old patients, the physicians described age to have a bigger influence. (Table 3, quote no. 1.3. and 1.4.)

Both patient age and functional capacity affected the assessment of quality of life with higher age and a perception of low functional capacity as indicators for poor quality of life. The physicians highlighted that time constraints increased the complexities in assessing the quality of life in patients unknown to them and mentioned the risk of attributing their own perception of quality of life to the patient. (Table 3, quote no. 1.5. and 1.6.)

**Expectations from surroundings.** The prehospital physicians described the expectations from relatives, bystanders and colleagues.

The physicians experienced that bystanders, such as first responders, expected them to continue the already-initiated resuscitation. These expectations were often not articulated, but merely implied. This led the physicians to continue treatment, in most cases for a limited time only, but long enough to show bystanders that "everything had been done" and that their

**Table 2. Themes, subthemes and overarching codes.**

| Themes | Subthemes | | Codes |
|---|---|---|---|
| Expectations towards and from parties involved in the cardiac arrest | Expectations about patient prognosis | | Information from the setting |
| | | | Expectations based on patient age |
| | | | Assessing quality of life |
| | Expectations from surroundings | | Relatives |
| | | | Bystanders |
| | | | Colleagues |
| The values and beliefs of the physicians and other involved parties | The patient's best interests and patient autonomy | | Avoiding unnecessary suffering |
| | | | Patient wishes |
| | | | Cardiac arrest is a merciful death |
| | | | Death is natural |
| | Equality | | Opinions on social value |
| | | | "What would I have wanted" |
| | Non-patient-related bioethical principles | | Involvement of relatives |
| | | | Relating to the patient or relatives |
| Dilemmas encountered in decision-making | Balancing conflicting expectations and values | | Treating comes with a risk of unnecessary suffering |
| | | | Pressure and threats from the relatives |
| | | | The greater good or the individual |
| | Doubts and uncertainty | | Incomplete or diverging information |

efforts had been worthwhile. (Table 3, quote no. 1.7.) Furthermore, some pointed to an overall societal expectation and perception that CPR always saves lives and felt that this could lead some bystanders to have unreasonably high expectations towards continued resuscitation. (Table 3, quote no. 1.8.) It was mentioned as important for the physicians to assess the relatives' expectations towards resuscitation as they experienced disagreements among the relatives as a complicating factor that could lead to continued resuscitation to avoid confrontations and potential complaints. (Table 3, quote no. 1.9.)

The physicians mentioned that they encountered expectations from prehospital or in-hospital colleagues. The prehospital colleagues were most often emergency medical technicians

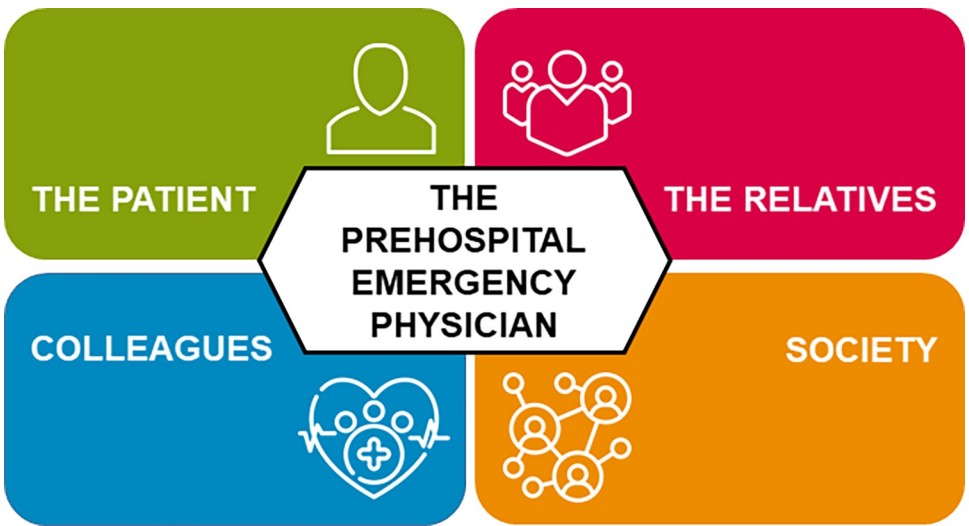

**Fig 2. Parties involved in decision-making.** A visualisation of the involved parties in OHCA decision-making.

(EMTs) who had initiated resuscitation before the physician had arrived at the scene and had preconceived expectations about continuation or termination depending on the patient and the crew. During one participant observation, an EMT became visibly upset because she believed the ongoing resuscitation attempt on an elderly nursing home resident was undignified and that the decision-making took too long. This created a conflict between the emergency physician and the EMT that was unresolved even after the termination of the resuscitation attempt. The physicians had experienced in-hospital personnel commenting openly on their decisions made prehospitally; most commonly the in-hospital personnel displayed frustration with what they believed to be futile resuscitation continued during the transport to the hospital. During one participant observation, the prehospital physician chose to continue resuscitation in the ambulance outside the hospital to achieve return of spontaneous circulation (ROSC) before presenting the patient to the in-hospital personnel because he did not feel he could "hand the patient over like that".

## The values and beliefs of the physicians and other involved parties

The physicians expressed a mixture of values and beliefs that all influenced the treatment. The values and beliefs included the patient's best interests, equality in treatment, and non-patient-related considerations.

**The patient's best interests and patient autonomy.** The physicians underlined the importance of avoiding unnecessary suffering for the patient. They were aware that should a

**Table 3. Table of quotes related to each theme.**

| Theme | No. | Quote | Participant no. |
|---|---|---|---|
| 1. Expectations towards and from parties involved in the cardiac arrest | 1.1. | And then you saw she had a care bed. And she had on one of those senior diapers. But she lived in her own home. You could see it was lovely and nice. Her glasses lay on the nightstand and the telephone was ready with a sort of. . . Was it a kind of beeper system?. . . Like a red thing. | P8 |
| | 1.2. | He was past his first youth, after all, you know? And then it matters. . . And it's difficult to say out loud but it plays a part that. . . Obviously, this was quite a miserable home, if we call a spade a spade, you know? | P10 |
| | 1.3. | If you're like 25 years old and you have ventricular fibrillation, ventricular fibrillation, and ventricular fibrillation. . . And then you have asystole or something like that. . . Then I would transport you to the hospital. Those are the ones that deserve a chance, in my opinion. On the other hand, if you're 98 and you have ventricular fibrillation that shifts to asystole and you don't respond after a while, then I would cease treatment. It probably is an influence. | P12 |
| | 1.4. | Well, there is a lot of information that we do not have. But the thing that means something to me as well is that we have a man who is 66 years old. And he can, of course, be terminally ill and all sorts, but it counts nonetheless. . . I adjust my mindset to push a little further than I normally would have if it were an 87-year-old. And that is age discrimination, but that's just how it is. | P13 |
| | 1.5. | And for some of them, they have to. . . They are bound to a wheelchair with a small cigarette holder and a tiny joystick. And in their opinion, it's a good life. Of course, they would prefer a different life, I don't doubt it, but they are pleased with what they got and they enjoy what they got, right? I can't be the judge of that when I'm out there, but then again, I am to some extent, right? | P9 |
| | 1.6. | And he's like completely cachectic, and you can tell that this is someone who is chronically bedridden. And then it's easier to make the decision than if it was someone who looked fairly good. You know she had on a senior diaper but a lot of elderly people have one of those. Of course, you can be incontinent but still have a decent quality of life right? | P8 |
| | 1.7. | If you imagine, that we had arrived out there all by ourselves, well then we had arrived at a dead person, right? But because it was a large-scale setup and a lot of things were happening, we arrived at a person with cardiac arrest who had resuscitation attempted. But he was dead, right? And that's the conclusion we had to reach without all the others finding it odd. In a way for them to. . . Yeah, we shouldn't ruin their day, you know. | P11 |
| | 1.8. | Well, the public gets a picture of getting resuscitated as first responders who come running with two jumper cables sticking them on your chest. Then it goes "Bang" and you're alive again. Then you'll sit up and eat the rest of your rye bread sandwich. That's just now how it works. I really miss somebody sharing the fact that if we succeed in restarting your heart, then your organ will have suffered. . . show signs of the lack of circulation or the cardiac arrest. That's why you have to be admitted to the ICU and be on a ventilator presumably for a longer period of time and with the risk of complications. I think it's a pity that image is not portrayed. | P16 |
| | 1.9. | But maybe others would say that's wrong, you know? And that's why I'm probing at the relatives because if they believe we're completely off the mark not to continue. . . Just all the nonsense caused by them being unsatisfied and complaining. They do that sometimes. . . You know, if they're very insisting, well then you'll do it anyway because then. . . Well then, alright, we'll transport them to the hospital, you know? | P8 |

*(Continued)*

**Table 3.** (Continued)

| Theme | No. | Quote | Participant no. |
|---|---|---|---|
| 2. The values and beliefs of the physicians and other involved parties | 2.1. | I can't stop thinking... Think if you had resuscitated that man, and he had been laying there for 20 minutes, and he just has to be admitted to the hospital and he has to incarcerate or something like that in a couple of days. I think that's unnecessary. It's unethical even. Because you know he'll never survive that. We don't have to resuscitate him just because we can. We also have to resuscitate, so that you have somewhat of a functional capacity, in my opinion. Especially if you're 63. Then you shouldn't lie... You just shouldn't lie as an empty vegetable. | P12 |
| | 2.2. | Well, I do think about the considerations you make before you begin treating a patient with cardiac arrest. Because it's an easy death, collapsing, and then you're gone if you have a chronic disease that results in not being able to survive anyway. It's a weary death if you're resuscitated too late, and your brain has said goodbye and thank you for everything. And then you come in here [the ICU] and have a treatment course... A heavy treatment course... | P15 |
| | 2.3. | I believe that most people wish to depart from this world by... Boom then it's over. I think very few people will say "Oh, I would really like a treatment course that drags out for 14 days where I'm put on a ventilator and have faecal catheter and food in a stomach tube and my relatives have to see me slowly fading away". I think fundamentally... My view of human nature is just not geared to believe that's the way people wish to depart from this world. Maybe some will say that, but I have a difficult time seeing that, and I have a hard time acting according to that. The thing about just collapsing... Boom-then-it's-over, it has to be the majority who dream about that. | P16 |
| | 2.4. | And then I assess... Well, we all have to die, and this woman just has to die now. Then... It's more about if... A person dies compared to somebody who suddenly suffers a cardiac arrest and has to be resuscitated. It's just that distinction, you know? | P1 |
| | 2.5. | Well, I believe the patient's autonomy to be inviolable. So if a patient does not want treatment, I believe that it should be respected. Well, then how should it be documented? Now we arrive at all the imaginary scenarios. Is it for the wife who now... Now she has all the life savings? But god dammit... If we have to run around being paranoid all the time, we should be semi-psychiatrists instead. I don't believe... I choose to believe that sometimes... In these situations as well, where people are under immense pressure, they are still in good faith. | P16 |
| | 2.6. | If a patient had said "I do not want to be resuscitated", as I said, he could have made the decision the day before yesterday and made a new decision yesterday but forgotten to register it. So there has to be some kind of... I have to be completely convinced that's how it is, and it's not a document the relatives have composed to get the inheritance into their own bank account right? | P15 |
| | 2.7. | I don't use it. Not at all. In my opinion, it's not... So far the legislation is a paradox in my opinion. Because you can't give up on yourself in advance. | P1 |
| | 2.8. | But it made everything easy. It was of a recent date. It was signed by the patient and stamped by the general practitioner even. "To whom it may concern. If I were to encounter this or this, I would under no circumstances..." Well, that cleared the table. That's the classic example. Because I don't have time to consult any electronic medical records, and we don't do that either. I don't at least. | P6 |
| | 2.9. | And once again, this is not the family man who functions well or... What's this story? Here is a worn-out person with a drug addiction that he doesn't necessarily have to die from, but now he *is* dead. Should you insist on saving that life at all costs? In that situation, it's easier to just do things. He responded very well [to treatment] as you saw. And I almost estimated his physiological age. You know, to the date. And then I don't believe... Well, you could see that he probably didn't have porridge for breakfast, but it wasn't... I don't believe the utility value, to use that ugly word in this context, I don't believe that disappeared. Whereas, if it's a skinny junkie, who really... You can tell that this is... Then it probably would have been something else. Or not probably. It *would* have been. | P6 |
| | 2.10. | And he is completely... He goes to work, and he has 25–30 years left of the job market. That's totally recouped. Then it's really worth it, you know? | P9 |
| | 2.11. | Well, of course, he is an addict, and of course, he has a sorry life but it doesn't write him off from having recovery potential. If we get him going and he makes it through intensive care. We have a tendency to, a lot of times, sentence these addicts to a permanently bad compliance. But some of them actually quits [drugs] in an emergency life event which I believe you can call this, and actually show compliance on the other side. And then they have the resources. They are still relatively young even though they have worn their bodies down. | P16 |
| | 2.12. | That's the experience I've had with infant deaths sometimes, you know? Where you just know that the child is dead, and you're performing CPR on a dead child. And you continue resuscitation till you arrive at the ED and you unload the parents in there and hand over [the patient], you know? And then you cease the treatment in there. Everybody agrees that the child is dead. But then you're at the hospital and the parents are at the hospital. There are professional grief counsellors, and somebody to take care of them. It's 100 times better being at the hospital in that situation. | P4 |
| | 2.13. | The circumstances in this situation are the wife is sitting in a car next to us and she is on the way to the hospital and she knows that we are continuing resuscitation. In my opinion, it's a bad time to just cease it all. When we arrive, we'll have to say "Well, he died in the ambulance on the way, you know?". Then I'll rather go up [to the catheterization laboratory] and then we can look each other in the eyes in cath lab and agree on what we can offer in this situation or what we can't offer. And then she's with him when he dies. So... Yeah. I believe she'll have more soul out of it in that way. Even if it's just for a short interval, right? Maybe it doesn't make that much of a difference, but... Then she's seen it herself. That everything has been done. | P4 |
| | 2.14. | It's probably because... Well, I don't bloody know... It's probably because we have children ourselves. And it seems unreasonable to take small children from someone. I think that's it. The child doesn't care. It's dead, you know. Thinking about the relatives having to... Yeah. How much they'll miss that child, makes you think:"Let's give it another chance". | P3 |
| | 2.15. | But if you have a young person with a family, then I'll think: "Maybe you'll be better off with cerebral sequelae because you know you can be part of your children's life" So it's a range of what do I believe and feel is reasonable? And that's of course based on my own life. So I think: "Would I rather have a hemiparesis and then be there for my children or..." Yes, I'd rather that than be given up on, and being told "There's no good coming out of this". Because what's quality of life? | P9 |

(*Continued*)

**Table 3.** (Continued)

| Theme | No. | Quote | Participant no. |
|---|---|---|---|
| 3. Dilemmas encountered in decision-making | 3.1. | Well, it's because sometimes it's. . . Well, sometimes it's nice because then. . . If it's a witnessed cardiac arrest and there's shockable rhythm and circulation, then it's fine. But if it's a non-witness cardiac arrest, non-shockable rhythm and they get some sort of stimulation, maybe epinephrine or something else, and they get circulation that you have to act on. . . That's an uncomfortable situation, right? Because you could say "How should I have avoided this?" Well, if I had. . . I should have decided faster. | P11 |
| | 3.2. | And then the EMTs initiate [CPR] as they should, and. . . uhm. . . then we're trapped. And I've tried that many times–being trapped. | P1 |
| | 3.3. | I'm okay with that. But then again it's mostly in cases where it's "young people anyway", who we'd go full throttle on and transport. I will be a part of that. I wouldn't have a problem with it. But an 86-year-old who has died in their own private surroundings, in my opinion. . . That's ugly. You have to be able to find peace, and I believe finding peace should be peaceful. | P16 |
| | 3.4. | If it's a young person, then. . . That person walking around without that heart or that liver, that kidney. . . It makes a tremendous difference. So if you have to do something good. . . If society could do something good for 2–5 people, right? In my opinion, that carries substantial weight. You really should prioritize that. | P13 |
| | 3.5. | But I do believe you would go the whole hog, and maybe even considering the circumstances. . . I could imagine crossing my own boundaries if it was children for instance. Also to show consideration for the relatives. But also consideration for myself and the personnel. For instance, it could be continuing futile resuscitation to get out of a, what to call it. . . a tense environment, if you follow? | P6 |
| | 3.6. | Well, it's probably mostly the discrepancy between what the relatives wish and hope for and what's technically possible. And you think to yourself:"Is it me who's completely in the wrong?" They have a huge demand for the patient to survive, and you're just thinking "But it makes no sense." | P7 |
| | 3.7. | I will say. . . Over the years, I've become more unaffected by the relatives' opinion, after I arrived at a 90-year-old's, and she had a somewhat retarded son, and she was the only one he had left. So he wished for her to have all possible treatment. And I thought "That's not up to him". But they tried to force us to initiate CPR on a really, really. . . you know a 90-plus-year-old lady. And if *I* think you shouldn't do it, then you shouldn't. I've also been in a situation, where we arrived at an Easter lunch and the husband had collapsed, and the wife screamed that he's very sick so we should stop [treatment]. But his two daughters and their husband is on the other side and tell us to do everything possible. | P12 |
| | 3.8. | Are they found in asystole, then that's it. Because it's. . . In my opinion, they have such a bad prognosis that I won't treat them further. . . But then again, right? But it's obvious if it's. . . you know, if it's a child or. . . if there is uncertainty about information. . . Then it's obvious, that barrier. . . How can you say. . . The barrier to doing something else has to be low. You know, in situations where there is doubt as I said before. | P10 |
| | 3.9. | But it's also. . . I believe you can make it tremendously easy if you want to. You just have to treat everybody. Then there aren't any considerations. Also, you could treat no one. That's a little harder because somebody would question that. But it's pretty easy like that as well. Because then you're always right. But in my opinion, the most important thing in cardiac arrest treatment is to walk the line, and not choose one side or the other. Continuously daring to battle yourself about what's the right thing to do? What's the decent thing to do? Then we can take 10 doctors and agree on disagreeing about what decency is. I don't think we'll ever get around that. But I think we need to quit the day we have chosen to be on only one side or the other. | P16 |

patient with non-favourable prognostic factors achieve ROSC, it would most likely lead to admission to the intensive care unit where the patient in turn would die. The physicians viewed this as unnecessary suffering and suffering because of the number of examinations and interventions the patient would have to endure in-hospitally. In such scenarios, physicians sometimes explicitly noted consideration for the patient's dignity in knowing when to terminate resuscitation in futile patients. (Table 3, quote no. 2.1.)

At the same time, the physicians indicated that they believed sudden cardiac arrest to be a merciful death for very old or chronically ill patients, describing it as an easy death where the patient's dignity, in most cases, was preserved. The in-hospital treatment was described as undignified compared to "being allowed to" die. (Table 3, quote no. 2.2. and 2.3.) The physicians noted that dying is natural and that "being born is a deadly disease". They highlighted that the distinction between "natural death" and "cardiac arrest" is essential in resuscitation decision-making. (Table 3, quote no. 2.4.)

The physicians mentioned assessing and including the patients' wishes concerning resuscitation in the decision process as a way of respecting the patient's autonomy by specifically looking for a DNACPR or by asking relatives for information on the patient's wishes. Some prehospital physicians described searching for the DNACPR only if they already had deemed the resuscitation attempt futile, and only including information about the patient's wishes that was relayed to them by the relatives or bystanders if it aligned with their clinical assessment.

Furthermore, the physicians' statements showed ambivalence towards the information and the wishes conveyed by the relatives. The physicians mentioned the relatives as a resource that provided important information about the patient's wishes in some cases, but also highlighted the risk of the relatives sharing misleading information to have their loved one receive maximum treatment or with ulterior motives such as inheritance. (Table 3, quote no. 2.5. and 2.6.)

Some physicians expressed similar ambivalence towards the use of DNACPR. Three physicians noted never using DNACPR, as they did not believe in the concept. Two of these physicians noted that the patient could have changed their mind since drafting the order. (Table 3, quote no. 2.7.) Among the remaining physicians, the opinions on advance directives were ambiguous describing the belief that the autonomy of the patient was inviolable but at the same time highlighting the difficulty in assessing the patient's wishes through DNACPR, because many patients did not have an advance directive. The lack of DNACPR was mentioned especially when treating nursing home residents where some physicians felt advance care planning should be mandatory. Two physicians noted an increasing number of DNACPR after the COVID-19 epidemic. The physicians noted having problems with accessing the Danish electronic register of living wills or expressed not having time to access the online registry. The physicians described the current Danish DNACPR form to be contradictory and difficult to utilize in practice as it was intricately verbose. (Table 3, quote no. 2.8.)

**Equality.** Equal treatment of patients was indirectly discussed in the physicians' statements. The physicians touched upon the "social value" of the patients and described some patients as being more important to rescue than others. Saving a patient with a job with many years of employment left or with children was mentioned as being more fulfilling than saving other patients, such as drug addicts or very old patients. Two physicians perceived drug addicts to have low compliance and as such a lower rehabilitation potential with one physician mentioning the patient's personality as being indicative of low rehabilitation potential. However, another physician underlined that drug addicts potentially could change their perspective on life after a life-threatening event. (Table 3, quote no. 2.9., 2.10. and 2.11.)

**Non-patient related considerations.** In the majority of observed cardiac arrests, the prehospital physicians were faced with relatives during and after the resuscitation. The physicians mentioned that they had continued the treatment, even to the point of transporting a patient to the hospital with ongoing cardiopulmonary resuscitation for the sake of the relatives. In some cases, this was to get the relatives to an in-hospital environment where they could receive bereavement counselling. This was most commonly mentioned in the treatment of children, even in cases where resuscitation was obviously futile. (Table 3, quote no. 2.12.)

The physicians mentioned continuing treatment either to show the relatives that everything had been done or to give the relatives time to say goodbye to the patient. During one participant observation, an elderly patient suffered cardiac arrest at home with his wife present. The patient was transported to the hospital with ongoing mechanical CPR with the wife sitting in a mobile emergency care unit that followed suit. During transportation, the physician conferred by telephone with in-hospital personnel who had consulted the patient's medical records. They advised the physician to terminate treatment due to severe comorbidities, but against this recommendation, the physician chose to continue treatment until arrival at the hospital. In the subsequent interview, the physician explained this: See Table 3, quote no. 2.13.

The physicians described relatives they could relate to influenced the decision-making process. Similarly, when physicians treated patients who had life situations comparable to their own, this affected them and made them reflect upon their own wishes in case of cardiac arrest. (Table 3, quote no. 2.14. and 2.15.)

### Dilemmas encountered in decision-making

The physicians experienced ethical dilemmas and challenges in the decision-making concerning resuscitation. These involved conflicting expectations and values, and doubt during the treatment.

**Balancing conflicting expectations and values.** The physicians experienced internal conflicts at the initiation and continuation of resuscitation due to the risk of unnecessary suffering. Most physicians had experienced initiating resuscitation while they gathered information from relatives, bystanders, or surroundings and by the time they felt they had enough information to terminate resuscitation, the patient had regained spontaneous circulation. This made them feel caught up in an obligation to continue resuscitation that, in reality, they did not feel was warranted. (Table 3, quote no. 3.1. and 3.2.) However, the physicians described the prospect of potential organ donation as a resulting benefit, but also highlighted the problem of treating one patient for the benefit of others. This reflection on prioritizing the greater good above the individual patient was made by a majority of physicians. Most physicians would always prioritize the patient in front of them, while a few reflected on how many people would benefit from one individual's organs. (Table 3, quote no. 3.3. and 3.4.)

The physicians described having encountered pressure from relatives to terminate resuscitation where the prehospital physician tended most towards the continuation of the treatment, but mostly to continue obviously futile resuscitation. Some physicians described violating their own boundaries to accommodate these wishes from the relatives, while others were explicit in not letting others influence their decision. (Table 3, quote no. 3.5.)

Some physicians described high expectations from relatives as leading to dilemmas. The discrepancy between what the physician believed to be clinically possible and what the relatives expected could make the physician question themselves and make the decision-making process complex. (Table 3, quote no. 3.6.) The wishes and expectations of the relatives could collide with each other and complicate decisions for the physicians. (Table 3, quote 3.7.)

**Doubts and uncertainty.** Almost all prehospital physicians mentioned the challenge of getting enough information to support the decision-making process. This challenge arose when relatives were not present or not able to provide information, but also when healthcare personnel present at the scene did not know the patient and thus could not provide information about either comorbidities or patient wishes. Most physicians described having continued resuscitation if they had had doubts about information or if they had lacked information, even though they might have had a gut feeling that resuscitation would not be successful. (Table 3, quote no. 3.8.) The physicians were aware of the self-fulfilling prophecy where they risked confirming their dismal prognoses on survival if they ceased resuscitation too early (Table 3, quote no. 3.9.).

## Discussion

Our results provide insight into the decision-making processes behind OHCA treatment. We found three overall themes describing the ethical aspects in prehospital emergency physicians' decision-making: Expectations towards treatment, the values and beliefs of the physicians and other involved parties, and dilemmas encountered in decision-making.

Some of these findings have also been reported in other studies on decision-making in out-of-hospital cardiac arrest. Expectations of poor survival and quality of life in old patients or patients with poor appearance may influence the prehospital personnel to terminate resuscitation [24]. Likewise, studies have described factors such as the location of the cardiac arrest, and bystander CPR to influence decision-making [25], as well as CPR before physician arrival

and advanced patient age among others [26]. Physicians' ethical considerations underlying the decision-making process have only been described in detail by a few studies [2].

Nordby et al. describe the specific challenges in the resuscitation of cancer patients and found that the participants considered it ethically correct to not resuscitate patients if they expected a short remaining survival, negative illness experiences, or a very low quality of life [27]. The physicians in our study underlined the difficulties in a prehospital setting in identifying patients with low quality of life and highlighted the risk of imposing one's own perception of quality of life upon the patients.

Our participants had several ways of assessing the patient's best interests and autonomy. One of these was advance directives or DNACPR. A systematic review concerning non-medical factors in prehospital resuscitation decision-making described discrepancies in the handling and inclusion of advance directives [2]. This corresponds with the findings in our present study, where some physicians as a matter of principle never included advance directives in the decision-making. Some physicians described only seeking advance directives if they had already made the decision to terminate resuscitation rendering the advance directive a tool to support their decision but not necessarily to further inform it. This finding supports quantitative studies, where patients with advance directives subjected to full resuscitation measures had favourable prognostic factors such as witnessed arrest, initial shockable rhythm, and ROSC obtained in the field [28]. This may be problematic as the patient's autonomy should be included and respected in decision-making.

The physicians in our study described the current form of the advance directive to be intricately verbose which made it difficult to use in practice. Some prehospital physicians described a lack of advance care planning in nursing homes and cases with very old patients as the cause of ethical dilemmas. This highlights the complexity of utilizing advance directives in the prehospital system and the importance of continuously evaluating implemented initiatives. In an extensive report from Perkins et al., the authors suggest a shift from isolated DNACPR decisions to wider integration of advance care planning in the overall care and treatment plans [29]. This strategy could be beneficial in a Danish context. By addressing the participants' concerns, challenges or problematic aspects of decision-making could be avoided.

Applying advance care planning in a wider context both medical and societal could address another challenge mentioned by the physicians in our study: Unrealistic expectations from the relatives regarding resuscitation. Engaging and educating patients and relatives before a cardiac arrest could decrease expectations and leave them prepared [3]. Distinguishing between cardiac arrest and end-of-life care by discussing advance care planning could facilitate more qualified decision-making. Studies describing low awareness and uptake rates even in developed countries underscore this as an area in need of improvement [30]. In line with our study, other qualitative studies describe the importance of having enough information to make the crucial decision to terminate resuscitation. In a model developed by Anderson et al. [31], the authors describe facilitators and challenges in decision-making and denote doubt, clinical concerns, emotional impact, and dismissive colleagues as challenges. The challenge that scarcity of information entails is difficult to solve because of the urgent and sudden nature of prehospital emergency medicine.

Social value was mentioned as potentially influencing decision-making. Social value in resuscitation has been described as an implicit social assessment that leads to putting more social value on some patients than on others [32]. Nurok et al. describe this phenomenon in a prehospital context, where age, sex, race, and socioeconomic status were part of the assessment of social value [32]. The prehospital physicians in our study mentioned treating younger patients more vigorously than older patients. They described feeling that the expectations from other healthcare professionals towards continuing the treatment were higher when

treating younger patients, while expectations towards terminating resuscitation were higher when treating the elderly. This could be a result of a perceived lower social value of the elderly, and a perception of having "more to save" in younger patients [32]. In an intensive care unit setting, Zussman argued that the difference between resuscitation efforts in the elderly and younger patients is due to the lower probability of survival in the elderly and not necessarily social value [33]. This may also be true in OHCA where some studies report each yearly increment in age decreased the probability of survival to hospital discharge by 2%. However, other studies report resuscitation of patients > 80 years to be worthwhile [34]. In our opinion, both perceptions of age were present in our study and they were intertwined to a degree that made it difficult to distinguish between them. The physicians both mentioned assessing the functional capacity as a proxy for the probability of successful resuscitation and rehabilitation, but at the same time, mentioned the value in saving "a family father" with many years left on the labour market. The physicians did not mention age as a factor in itself except for children or the very old. In OHCA treatment, there is a risk of confirming a self-fulfilling prophecy, as old age in itself may not be associated with a poor outcome [34]. As such, old age should not be used as a singular factor in decision-making. Instead, studies suggest that frailty may predict outcomes [35]. The physicians in our study did describe assessing frailty often using subjective perceptions.

Socioeconomic status was mentioned by our participants. They assessed the status of the patient's home and some participants perceived that patients with low socioeconomic status would have a lack of compliance regarding any treatment following the cardiac arrest. Current evidence suggests a socioeconomic and racial inequality in the entire chain of survival in OHCA [36], but also in end-of-life care [37] where higher socioeconomic status also entails more advance care planning. This potentially leaves a gap where patients with low socioeconomic status could receive suboptimal OHCA treatment and be bereaved of proper end-of-life care. The physicians in our study described being less willing to continue resuscitation in patients with drug addictions while being more willing to continue resuscitation in patients who had a life situation similar to their own based on prima facie perceptions of susceptibility for rehabilitation and expected level of compliance. In our opinion, decision-making processes in cardiac arrest may be yet another factor that in some instances enhances the social inequalities in healthcare.

Being continuously exposed to these challenging ethical dilemmas may lead to moral distress in prehospital clinicians [38]. In the Danish prehospital setting, no official organisational debriefing processes or other initiatives to manage these challenges currently exist [39]. Initiatives to reduce the risk of mortal distress in psychiatric and health community settings involve ethical reflection groups where healthcare professionals engage in transparent and structured discussions and evaluations of current or previous ethical dilemmas [40]. Such structured approaches have not been developed or tested in the prehospital setting but are warranted. Likewise, training involving ethical aspects or challenges has been proposed within the in-hospital setting [39,41,42], but similarly has not been attempted in the prehospital setting where the organisational structure and work environment may impact how and when training is feasible.

Some areas of medical decision-making and influencing ethical considerations described in this study may be attributed to cognitive biases, which are well-known in medical decision-making [43]. Studies on how to reduce the influence of cognitive biases suggest decision-making tools or the use of technology e.g. electronic decision-making aids [43]. These suggestions may not be feasible in ethical decision-making where each ethical dilemma is highly contextual and therefore cannot be standardised to fit a template [44]. However, some challenges involving decision-making and DNACPR could potentially be solved by integrating DNACPR

orders into the existing prehospital electronic medical records system. Other suggestions such as reflective practice where clinicians actively reflect on decision-making could potentially reduce cognitive biases [43]. This may include ethical reflection groups which have been used in other contexts such as psychiatric healthcare [40]. The reflective practice allows for doubts, beliefs, and values to be discussed with transparency and may increase awareness of each individual's own biases [40]. Similarly, education in ethical practice such as ethics competence learning may improve ethical decision-making [45,46]. Both education and reflective practices could potentially reduce the risks of cognitive biases but also minimise the risk of moral distress.

Overall, improvements in decision-making cannot be reduced to simple guidelines or regulations but require wider initiatives involving interventions aiming at several parts of the healthcare system [40]. These initiatives require the involvement of patients, relatives, and the public at large to educate and inform about cardiac arrest and advance care planning [3]. As an example, the number of ethical challenges concerning DNACPRs could be reduced by interdisciplinary initiatives involving clinicians from both the prehospital, in-hospital, and community health sectors. By raising awareness among the physicians seeing the patient before a potential cardiac arrest, decision-making on DNACPR in a non-acute setting could be facilitated and the number of ethical challenges potentially be minimised [47].

Our findings add to the knowledge surrounding resuscitation and, together with quantitative studies on resuscitation, provide new perspectives on decision-making. An important finding is the complexity of OHCA decision-making. Each theme presented in this study could be further researched in-depth, but it is important to remember the holistic approach to OHCA decision-making as many aspects are interrelated. This study provides underlying reasons for variations in decision-making and potentially in survival due to decision-makers' biases and differences in the weighting of ethical considerations.

## Limitations

One limitation is that conclusions made based on the Danish EMS that includes prehospital anaesthesiologists may not be generalizable to other prehospital systems devoid of physicians. Another limitation is that the observer/interviewer, author LM, is a physician. This may have influenced the interviews due to insider knowledge and differences in preunderstandings compared to outsiders. Insider knowledge can facilitate transparency and honesty that would not have been achieved by an outsider [18]. By having a reflective approach induced by keeping a field diary, by a continuous discussion with the research team, and by the involvement of a variety of researchers in the analysis and interpretation, we sought to minimize the impact of these preunderstandings. Participant observations carry a risk of bias in the form of the Hawthorne effect where the observed participants alter their behaviour because they know that they are observed. Furthermore, post hoc self-justification by the attending prehospital anaesthesiologists may have influenced the interviews just as recall bias could have impacted the themes mentioned. By conducting naturalistic participant observations where we observed daily life and interactions we sought to reduce these risks.

## Conclusion

In this multicentre qualitative study, we examined real-life ethical decision-making during prehospital OHCA treatment. The physicians described a wide array of ethical aspects in decision-making in OHCA. Expectations from others towards the physicians and from the physicians toward others both facilitated and challenged decision-making. The physicians' and the surroundings' values and beliefs influenced decision-making, and challenges and dilemmas

arose when values, beliefs, and expectations from physicians, patients, relatives, and colleagues collided with each other. Several aspects of ethical decision-making were identified that could be harmful to both physicians and patients such as unequal treatment based on subjective evaluations of factors like socioeconomic status and variations in the inclusion and interpretation of DNACPR. Multifaceted interventions on a wider societal level like integration and evaluation of advance care planning in different healthcare sectors could facilitate decision-making. Future research should focus on potential interventions to safeguard quality in ethical decision-making in prehospital resuscitation to minimize inequalities.

## Supporting information

**S1 File. COREQ guideline.**
(PDF)

**S2 File. Translated interview guide.**
(PDF)

**S3 File. Translated observation and field note guide.**
(PDF)

## Acknowledgments

Thank you to the OPEN Network at OUH for advice and assistance with data management. Thanks to all participants and the health regions for their participation and willingness to be part of the study.

## Author Contributions

**Conceptualization:** Louise Milling, Dorthe Susanne Nielsen, Jeannett Kjær, Lars Grassmé Binderup, Caroline Schaffalitzky de Muckadell, Helle Collatz Christensen, Erika Frischknecht Christensen, Annmarie Touborg Lassen, Søren Mikkelsen.

**Data curation:** Dorthe Susanne Nielsen.

**Formal analysis:** Louise Milling, Dorthe Susanne Nielsen.

**Funding acquisition:** Louise Milling.

**Investigation:** Louise Milling, Søren Mikkelsen.

**Methodology:** Louise Milling, Dorthe Susanne Nielsen, Jeannett Kjær, Lars Grassmé Binderup, Caroline Schaffalitzky de Muckadell, Annmarie Touborg Lassen, Søren Mikkelsen.

**Project administration:** Louise Milling, Jeannett Kjær, Annmarie Touborg Lassen, Søren Mikkelsen.

**Software:** Louise Milling.

**Supervision:** Dorthe Susanne Nielsen, Lars Grassmé Binderup, Caroline Schaffalitzky de Muckadell, Helle Collatz Christensen, Erika Frischknecht Christensen, Annmarie Touborg Lassen, Søren Mikkelsen.

**Validation:** Dorthe Susanne Nielsen.

**Visualization:** Louise Milling.

**Writing – original draft:** Louise Milling.

**Writing – review & editing:** Louise Milling, Dorthe Susanne Nielsen, Jeannett Kjær, Lars Grassmé Binderup, Caroline Schaffalitzky de Muckadell, Helle Collatz Christensen, Erika Frischknecht Christensen, Annmarie Touborg Lassen, Søren Mikkelsen.

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
