## [Decision Letter · Decision Letter 0]

13 Feb 2023

PONE-D-22-34819Ethical considerations in the prehospital treatment of out-of-hospital cardiac arrest: a multi-centre, qualitative studyPLOS ONE

Dear Dr. Milling,

Thank you for submitting your manuscript to PLOS ONE. After careful consideration, we feel that it has merit but does not fully meet PLOS ONE’s publication criteria as it currently stands. Therefore, we invite you to submit a revised version of the manuscript that addresses the points raised during the review process. Please submit your revised manuscript by Mar 30 2023 11:59PM. If you will need more time than this to complete your revisions, please reply to this message or contact the journal office at plosone@plos.org. Please include the following items when submitting your revised manuscript:A rebuttal letter that responds to each point raised by the academic editor and reviewer(s). You should upload this letter as a separate file labeled 'Response to Reviewers'.A marked-up copy of your manuscript that highlights changes made to the original version. You should upload this as a separate file labeled 'Revised Manuscript with Track Changes'.An unmarked version of your revised paper without tracked changes. You should upload this as a separate file labeled 'Manuscript'.

We look forward to receiving your revised manuscript.

Kind regards,

Federica Canzan

Academic Editor

PLOS ONE

Journal Requirements:

Reviewers' comments:

Reviewer's Responses to Questions

**Comments to the Author**

1. Is the manuscript technically sound, and do the data support the conclusions?

Reviewer #1: Yes

Reviewer #2: Partly

2. Has the statistical analysis been performed appropriately and rigorously? 

Reviewer #1: N/A

Reviewer #2: N/A

3. Have the authors made all data underlying the findings in their manuscript fully available?

Reviewer #1: No

Reviewer #2: Yes

4. Is the manuscript presented in an intelligible fashion and written in standard English?

Reviewer #1: Yes

Reviewer #2: Yes

5. Review Comments to the Author

Reviewer #1: An interesting and potentially important contribution to the limited literature available on complex decision making by clinicians in the course of OHCA. The qualitative methodology provides an effective mechanism to explore decision making in ways which are more commonly enumerated, thus limiting insight into the difficulties faced by clinicians. Issues for clarification;

1. The 'observation guide' and 'field notes' instruments are not described or included. Both are relevant and should be included in an appendix.

2. No research ethics approval has been sought or granted for this study, based on the authors view that Danish law does not require approval in such studies. This contention will have to be supported, as it seems unlikely that observational, qualitative research of this nature is immune from ethical considerations. The study otherwise appears to conform rigorously to the approaches required for this work

3. The presence of the researcher on scene during the OHCAs is described as 'author LM balanced between moderate and active participation alternating between the insider and outsider perspectives'. Please explain what this means, the extent of your involvement in individual cases and the potential for you to have influenced participants or case management or to no longer be regarded as an independent researcher by the study subjects. This is a particularly important issue given the qualitative format of the data collection following each incident.

4. How many interviews were carried out, relating to how many OHCAs?

5. The thematic analysis identifies interesting and important issues, well illustrated with appropriate quotes, which will resonate with clinicians involved in such care. Many observations are personal but raise issues such as the extent to which doctors believe their own values should dictate care, the value of DNACPR ' advance directives or the real-work experience of team working. While the discussion summarises these and other issues, it does little to explore their meaning within the broader healthcare system or the implications for e.g. training, discussion with family members (or patients considering DNACPR orders) or if more collaborative decision making on scene has a role.

5. Is there an operational 'learning process' in place for doctors, paramedics etc following these cases e.g. hot or cold debriefing, lessons learned, system learning issues, case reviews, CISM etc? Burnout among clinicians exposed to these stressors on a regular basis has to be more likely if their frustrations e.g. with nursing home advance care directives are not seen to be addressed. An implication from the description of the EMS system is that doctors and paramedics train separately and meet only at clinical incidents - is training an issue to explore in the discussion.

6. Some reflection on future research steps is contained in the discussion - the authors might consider expanding on this in view of some of the issues above.

7. There are a small number of typos in the text - full proofing required. Would suggest changing 'end-of-life' to 'end-of-life care' where relevant.

Reviewer #2: I read this paper with interest. However, I would require greater clarity on the qualitative methods employed in this study, moreover, there are gaps in the link between the theoretical framework employed, study methods, and study findings.

Specific comments:

1. "By comparison, out-of-hospital resuscitation is characterized by time pressure and a lack of peer consultation and background information" - I am not sure the ethical dilemmas implied by the authors are well articulated. When reading the introduction, the sense I get is in fact that paramedics should just commence resuscitation when in doubt, especially given the lack of information and time-sensitive nature of out-of-hospital cardiac arrests. Most countries also have Good Samaritan laws that provide protection against liability for paramedics, EMTs and bystanders at the scene of an emergency or while en route to a hospital from an emergency, they will not be held legally responsible for acting or failing to act. The introduction section is also too brief at the moment. I suggest the authors give more concrete examples of the ethical dilemmas in the introduction to provide context to their study.

2. How far apart were the six different MECU bases?

3. Given the centrality of the theory of bioethics in resuscitation as a theoretical framework for analysing the study findings, this should also have been laid out in the introduction section for readers to appreciate the context of the study. Why is the aforementioned theory an appropriate theoretical frame to understand your chosen data source?

4. In terms of the methods, what type of specific qualitative method did the authors employ? What was the coding cycle? How did you arrive at themes? What was the intercoder reliability in terms of deciding which topics related to which theme?

5. I am not really following the themes of the study. When reading the verbatims, I get the feeling that age of the person in arrest is a more critical influencing factor and this could stem from ageism bias (which is not uncommon in healthcare when have mental idea or explicit cutoffs for treatment or resource allocation), perceived notion that death of a young and otherwise healthy individual is unnatural versus very old person dying from natural, age-related causes, and overall perceived burden on society (biomedical/ethical principle of justice and equity). All these quotes demonstrate this idea, e.g. "But then again it’s mostly in cases where it’s “young people anyway”, who we’d go full throttle on and transport. I will be a part of that. I wouldn’t have a problem with it", "If you’re like 25 years old and you have ventricular fibrillation, ventricular fibrillation, and ventricular fibrillation… And then you have asystole or something like that… Then I would transport you to the hospital" and "He goes to work, and he has 25-30 years left of the job market. That’s totally recouped", but they were classified under different themes. I am not really following or convinced by the analysis presented by the paper.

6. It may be more straightforward to categorise your themes according to the four basic tenets of biomedical ethics. Bioethicists often refer to the four basic principles of health care ethics when evaluating the merits and difficulties of medical procedures. Ideally, for a medical practice to be considered "ethical", it must respect all four of these principles: autonomy, justice, beneficence, and non-maleficence.

7. "Expectations of poor survival and quality of life in old patients ..." - are these expectations not reasonable? Even when EMS was activated promptly, the rate of survival to discharge among EMS-treated OHCA patients was approximately only 10% (citation: pubmed.ncbi.nlm.nih.gov/25676321). Given that OHCAs have poor survival as a whole, and that multivariate analysis has demonstrated that each yearly increment in age decreased the probability of survival to hospital discharge by 2% (citation: ncbi.nlm.nih.gov/pmc/articles/PMC5805233), it is important to have a more robust discussion about the implicit and explicit issues herein.

8. "...studies have described factors such as the location of the cardiac arrest" - it is important to mention that arrest in residential settings and high-rise buildings have been invariably associated with poorer outcomes (citation: ncbi.nlm.nih.gov/pmc/articles/PMC8539960).

9. "Applying advance care planning in a wider context both medical and societal could address another challenge mentioned by the physicians in our study" - it is important to mention the caveat that the awareness and uptake rates of ACP have been low, even in developed countries (citation: pubmed.ncbi.nlm.nih.gov/28417132).

10. No discussion of ethics is complete without an accompanying discussion of the legal framework. Based on my understanding of the Danish law, under the Danish penal code, "all persons must provide aid to the best of their ability to any person who appears to be lifeless or in mortal danger, must alert authorities or take similar steps to prevent impending disasters that could cause loss of life, must comply with all reasonable requests of assistance by a public authority when a person's life, health or well-being is at stake, and must, if they learn of a planned crime against the state, human life or well-being, or significant public goods, do everything in their power to prevent or mitigate the crime, including but not limited to reporting it to authorities, in all cases provided that acting would not incur particular danger or personal sacrifice." Would this not influence the decision-making of first-responders given the law compels them to act?

6. PLOS authors have the option to publish the peer review history of their article (what does this mean?). If published, this will include your full peer review and any attached files.

Reviewer #1: **Yes: **Gerard Bury

Reviewer #2: No

---

## [Author Response · Author response to Decision Letter 0]

31 Mar 2023

Reviewer #1: An interesting and potentially important contribution to the limited literature available on complex decision making by clinicians in the course of OHCA. The qualitative methodology provides an effective mechanism to explore decision making in ways which are more commonly enumerated, thus limiting insight into the difficulties faced by clinicians. Issues for clarification;

1. The 'observation guide' and 'field notes' instruments are not described or included. Both are relevant and should be included in an appendix.

Authors’ response: Thank you for an excellent suggestion. We have translated the observation and field note guide and included it as Appendix 3.

2. No research ethics approval has been sought or granted for this study, based on the authors view that Danish law does not require approval in such studies. This contention will have to be supported, as it seems unlikely that observational, qualitative research of this nature is immune from ethical considerations. The study otherwise appears to conform rigorously to the approaches required for this work

Authors’ response: This is indeed a relevant consideration. In the present study, the study objectives were the prehospital physicians’ decision-making and their decision drivers. All prehospital physicians were informed of the purpose of the study and all had the option of opting-out of their specific participation before, during, and after the observations and the interviews. To address the reviewer’s concerns, we have contacted The Regional Committees on Health Research Ethics for Southern Denmark to discuss any post-hoc ethical approval of the study, and have received the following response: 

“The Regional Committees on Health Research Ethics for Southern Denmark have received your request on whether your research project is subject to notification under the Committee Act. Your request has been given case number 20232000 – 23. Based on the available information, the Committee assessed that the project could not be considered health science research with an obligation to report to the Health Research Ethics Committee System, as found in Committee Act Article §14, paragraph 1. In the decision, emphasis has been placed on the fact that it appears to be a project where the trial-related procedures cannot be regarded as an intervention within the meaning of the Committee Act. The project, therefore, falls outside the scope of the Committee Act's definition of a reportable health science research project. Pursuant to the Committee Act Article §14, paragraph 2, questionnaire surveys and register-based research projects must only be reported to the Research ethics committee system if the project includes human biological material. According to the committee system's notification instructions, this also applies to interview surveys: https://www.nvk.dk/forsker/naar-du-anmelder/hvilke-projekter-skal-jeg-anmelde [in Danish]. If a health research project is to fall within the framework for reporting to the committee system, the project must, as found in the guidelines from the National Committee on Health Research Ethics, both have a health scientific purpose and involve intervention: http://www.nvk.dk/forsker/naar-du-anmelder/hvilke-projekter-skal-jeg-anmelde. Your request was assessed by the Chairman of the Regional Committee 2, chief physician Aia Elise Jønch, PhD.”

We have added the following paragraph in the manuscript to clarify this p. 6, l. 159-160: “The study was assessed by The Regional Committees on Health Research Ethics for Southern Denmark (no. 20232000 – 23) and exempted from ethical approval”.

3. The presence of the researcher on scene during the OHCAs is described as 'author LM balanced between moderate and active participation alternating between the insider and outsider perspectives'. Please explain what this means, the extent of your involvement in individual cases and the potential for you to have influenced participants or case management or to no longer be regarded as an independent researcher by the study subjects. This is a particularly important issue given the qualitative format of the data collection following each incident.

Authors’ response: Spradley describes the moderate participant as “watcher” from the outside never really gaining the skill or status as the observed, whereas the active participant seeks to do what other people are doing to gain acceptance and learn the social rules. During observations, the author LM at times became a part of the daily life of the prehospital physicians and the team of EMTs and paramedics. This meant engaging in tasks such as the daily check of the MECU and assist in carrying bags or equipment shifting the participation from moderate to active. 

The insider and outsider perspectives are described by Spradley as natural roles an observer will fall into during participant observations. The insider will experience situations in an immediate subjective manner, while outsiders will observe the situation from the outside. As Spradley describes, participant observers will usually shift between the two perspectives continuously and at times be able to attain both perspectives. This was also the case in this study. 

We have added these descriptions of participant observations to the Method section p. 5, l.119-129: “Spradley describes the moderate participant as a “watcher” from the outside never really gaining the same skill or status as the observed, whereas the active participant seeks to do what other people are doing to gain acceptance and learn the social rules (18). During observations, the author LM at times became a part of the daily life of the prehospital physicians and the team of EMTs and paramedics. This meant engaging in tasks such as the daily check of the MECU and help carrying bags or equipment shifting the participation from moderate to active (18). Likewise, the insider and outsider perspectives are described as natural roles an observer will fall into during participant observations (18). The insider will experience situations in an immediate subjective manner, while outsiders will observe the situation from the outside. Participant observers will usually shift between the two perspectives continuously and at times be able to attain both perspectives (18). This was also the case in this study.”

4. How many interviews were carried out, relating to how many OHCAs?

Authors’ response: This has been specified in the “Methods” section, page 5, l. 107-108 with the sentence: “We observed 22 prehospital cardiac arrest treatments in total, and subsequently interviewed the 17 physicians who were responsible for the treatment in each case”. 

5. The thematic analysis identifies interesting and important issues, well illustrated with appropriate quotes, which will resonate with clinicians involved in such care. Many observations are personal but raise issues such as the extent to which doctors believe their own values should dictate care, the value of DNACPR ' advance directives or the real-work experience of team working. While the discussion summarises these and other issues, it does little to explore their meaning within the broader healthcare system or the implications for e.g. training, discussion with family members (or patients considering DNACPR orders) or if more collaborative decision making on scene has a role.

Authors’ response: This is a very important perspective – especially in the light of the findings that termination of resuscitation is based largely on the individual physician’s perceptions at the scene. It underscores our general notion that education is indeed required in order to increase the awareness among the prehospital physicians of this intangible area. We have aimed to emphasise this by adding a section on implications in a broader healthcare perspective. This section has now been embedded into the “Discussion” section p. 16, l. 438-461: “Some areas of medical decision-making and influencing ethical considerations described in this study may be attributed to cognitive biases, which are well-known in medical decision-making (42). Studies on how to reduce the influence of cognitive biases suggest that decision-making tools or the use of technology e.g. electronic decision-making aids should be implemented (42). These suggestions may not be feasible in ethical decision-making where each ethical dilemma is highly contextual and therefore cannot be standardised to fit a template (43). However, some challenges involving decision-making and DNACPR could potentially be solved by integrating DNACPR orders into the existing prehospital electronic medical records system. Other suggestions such as reflective practice where clinicians actively reflect on decision-making could potentially reduce cognitive biases (42). This may include ethical reflection groups, which have been used in other contexts such as psychiatric healthcare (38). The reflective practice allows for doubts, beliefs, and values to be discussed with transparency and may increase awareness of each individual’s own biases (38). Similarly, education in ethical practice such as ethics competence learning may improve ethical decision-making (44, 45). Both education and reflective practices could potentially reduce the risks of cognitive biases but also minimise the risk of moral distress. Overall, improvements in decision-making cannot be reduced to simple guidelines or regulations but require wider initiatives involving interventions aiming at several parts of the healthcare system. These initiatives require the involvement of patients, relatives, and the public at large to educate and inform about cardiac arrest and advance care planning (3). As an example, the number of ethical challenges concerning DNACPRs could be reduced by interdisciplinary initiatives involving clinicians from both the prehospital, in-hospital, and community health sectors. By raising awareness among the physicians seeing the patient before a potential cardiac arrest, decision-making on DNACPR in a non-acute setting could be facilitated and the number of ethical challenges potentially be minimised (46). ” 

5 Is there an operational 'learning process' in place for doctors, paramedics etc. following these cases e.g. hot or cold debriefing, lessons learned, system learning issues, case reviews, CISM etc.? Burnout among clinicians exposed to these stressors on a regular basis has to be more likely if their frustrations e.g. with nursing home advance care directives are not seen to be addressed. An implication from the description of the EMS system is that doctors and paramedics train separately and meet only at clinical incidents - is training an issue to explore in the discussion.

Authors’ response: Thank you for addressing this very important issue. There are not any official, organisational debriefing programs in the Danish EMS system. Some local EMS bases employed their own case review sessions and debriefing procedures, but the lack of structured overall initiatives to address this is indeed an issue that should be discussed. We have added the following paragraph in the discussion addressing the topic p.16, l.427-436: “Being continuously exposed to these challenging ethical dilemmas may lead to moral distress in prehospital clinicians (37). In the Danish prehospital setting, no official organisational debriefing processes or other initiatives to manage these challenges currently exist. Initiatives to reduce the risk of mortal distress in psychiatric and health community settings involve ethical reflection groups where healthcare professionals engage in transparent and structured discussions and evaluations of current or previous ethical dilemmas (38). Such structured approaches have not been developed or tested in the prehospital setting but are warranted. Likewise, training involving ethical aspects or challenges has been proposed within the in-hospital setting (39-41), but similarly has not been attempted in the prehospital setting where the organisational structure and work environment may impact how and when training is feasible.”

6. Some reflection on future research steps is contained in the discussion - the authors might consider expanding on this in view of some of the issues above.

Authors’ response: Please see our response to question #4. 

7. There are a small number of typos in the text - full proofing required. Would suggest changing 'end-of-life' to 'end-of-life care' where relevant.

Authors’ response: This is indeed embarrassing. We have scrutinised the manuscript and sought to eliminate these errors, just as we have changed the wording 'end-of-life' to 'end-of-life care' where relevant.

Reviewer #2: I read this paper with interest. However, I would require greater clarity on the qualitative methods employed in this study, moreover, there are gaps in the link between the theoretical framework employed, study methods, and study findings.

Specific comments:

1. "By comparison, out-of-hospital resuscitation is characterized by time pressure and a lack of peer consultation and background information" - I am not sure the ethical dilemmas implied by the authors are well articulated. When reading the introduction, the sense I get is in fact that paramedics should just commence resuscitation when in doubt, especially given the lack of information and time-sensitive nature of out-of-hospital cardiac arrests. Most countries also have Good Samaritan laws that provide protection against liability for paramedics, EMTs and bystanders at the scene of an emergency or while en route to a hospital from an emergency, they will not be held legally responsible for acting or failing to act. The introduction section is also too brief at the moment. I suggest the authors give more concrete examples of the ethical dilemmas in the introduction to provide context to their study.

Authors’ response: Thank you for this comment. In cases where the paramedics are the sole decision makers at the scene, the points made by this reviewer are fully relevant. However, as described in the section “System setting”, the decision to terminate the treatment of a patient with out-of-hospital cardiac arrest when there are no obvious signs of death (decapitation, decomposition) is always made by a physician. This decision in itself requires ethical considerations as cases may be considered futile even without these obvious signs of death. The present study aims to investigate this. We have elaborated the differences between situations where a paramedic – according to legislation – is required to continue resuscitation and a situation where a physician is required to make a medically based decision to terminate the treatment in a situation considered futile or ethically challenging.

2. How far apart were the six different MECU bases?

Authors’ response: The MECU bases are located throughout Denmark with approximately 100-200 km between each base, and 400 km between the two bases furthest apart. Denmark is a small country and the longest distance across the country is 500 km. We have added this in the method section p. 4, l. 103-105.

3. Given the centrality of the theory of bioethics in resuscitation as a theoretical framework for analysing the study findings, this should also have been laid out in the introduction section for readers to appreciate the context of the study. Why is the aforementioned theory an appropriate theoretical frame to understand your chosen data source?

Authors’ response: Thank you for the relevant suggestion. We have added a paragraph in the introduction on bioethics including a specification of why this framework is appropriate in this study on p. 3, l. 66-73: “Ethical considerations are a universal part of medical decision-making and have been described in both somatic, psychiatric, and health community settings (7-10). The four bioethical principles proposed by Beauchamp and Childress are acknowledged as important parts in the treatment of OHCA in the European Resuscitation Guidelines (3). They “1) Respect for autonomy (respecting the decision-making capacities of autonomous persons), Non-maleficence (avoid causing harm), 3) Beneficence (provide benefits and to balance benefits against risks) and 4) Justice (obligations of fairness in the distribution of benefits and risks)" (11). These four principles (12) are universal, can be applied in all healthcare settings (13) and should guide decision-making for healthcare professionals (3).”

4. In terms of the methods, what type of specific qualitative method did the authors employ? What was the coding cycle? How did you arrive at themes? What was the intercoder reliability in terms of deciding which topics related to which theme?

Authors’ response: As described in the “Methods”-section, we employed an ethnographic study using a hermeneutic-phenomenological approach. In the data analysis, we employed a thematic analysis using an inductive-deductive method, where previous literature informed the coding, while we simultaneously remained open in the discovery of codes and themes. Since, we have conducted this study according to the principles of qualitative research where intercoder reliability is not assessed nor viewed as a valid measurement to assess the validity of the results. In qualitative research, the aim is to elaborate themes and elucidate phenomena through reflective analysis, and not investigate the distribution or frequencies of meanings and perceptions in the respondents. 

5. I am not really following the themes of the study. When reading the verbatims, I get the feeling that age of the person in arrest is a more critical influencing factor and this could stem from ageism bias (which is not uncommon in healthcare when have mental idea or explicit cutoffs for treatment or resource allocation), perceived notion that death of a young and otherwise healthy individual is unnatural versus very old person dying from natural, age-related causes, and overall perceived burden on society (biomedical/ethical principle of justice and equity). All these quotes demonstrate this idea, e.g. "But then again it’s mostly in cases where it’s “young people anyway”, who we’d go full throttle on and transport. I will be a part of that. I wouldn’t have a problem with it", "If you’re like 25 years old and you have ventricular fibrillation, ventricular fibrillation, and ventricular fibrillation… And then you have asystole or something like that… Then I would transport you to the hospital" and "He goes to work, and he has 25-30 years left of the job market. That’s totally recouped", but they were classified under different themes. I am not really following or convinced by the analysis presented by the paper.

Authors’ response: Thank you very much for the suggestions. Age as such is definitely an issue in our analysis. However, age did not stand out as a single factor but rather as one factor among others. This is why we have chosen to present this particular finding as we have done. In our analysis process, where all authors were involved, we chose to include age in all themes because it was a factor informing the ethical decision-making but not defining it. 

6. It may be more straightforward to categorise your themes according to the four basic tenets of biomedical ethics. Bioethicists often refer to the four basic principles of health care ethics when evaluating the merits and difficulties of medical procedures. Ideally, for a medical practice to be considered "ethical", it must respect all four of these principles: autonomy, justice, beneficence, and non-maleficence.

Authors’ response: Thank you very much for this suggestion. As we chose a thematic analysis with a phenomenological hermeneutic approach, defining themes before engaging in analysis would not follow the methodology. Likewise, the hermeneutic approach aims towards finding and describing new meanings and not confirming previous. As such, we were not able to categorise the questions beforehand nor the responses post-hoc. This was done to be true to the methodology.

7. "Expectations of poor survival and quality of life in old patients ..." - are these expectations not reasonable? Even when EMS was activated promptly, the rate of survival to discharge among EMS-treated OHCA patients was approximately only 10% (citation: pubmed.ncbi.nlm.nih.gov/25676321). Given that OHCAs have poor survival as a whole, and that multivariate analysis has demonstrated that each yearly increment in age decreased the probability of survival to hospital discharge by 2% (citation: ncbi.nlm.nih.gov/pmc/articles/PMC5805233), it is important to have a more robust discussion about the implicit and explicit issues herein.

Authors’ response: Thank you very much for highlighting this important issue. We have elaborated on the topic of survival in the elderly in p.15, l.407-409. “This may also be true in OHCA where some studies report each yearly increment in age decreased the probability of survival to hospital discharge by 2%. However, other studies report resuscitation of patients > 80 years to be worthwhile (34). In our opinion, both perceptions of age…”. We have not attempted to judge the responses given by the respondents, merely report them. 

8. "...studies have described factors such as the location of the cardiac arrest" - it is important to mention that arrest in residential settings and high-rise buildings have been invariably associated with poorer outcomes (citation: ncbi.nlm.nih.gov/pmc/articles/PMC8539960).

Authors’ response: Thank you for addressing this aspect of prognostic factors in OHCA. However, in the Danish context, this is not an issue of relevance. Denmark has very few high-rise buildings, and the participants did not mention this as an issue. Furthermore, the sentence the reviewer refers to is related to EMS clinicians’ decision-making and not prognostic outcomes. This issue should be addressed in future studies where the setting and context encompass this aspect, but we consider it out of scope for this study. 

9. "Applying advance care planning in a wider context both medical and societal could address another challenge mentioned by the physicians in our study" - it is important to mention the caveat that the awareness and uptake rates of ACP have been low, even in developed countries (citation: pubmed.ncbi.nlm.nih.gov/28417132).

Authors’ response: We have added the comment p.15, l.389-390: “Studies describing low awareness and uptake rates even in developed countries underscores this as an area in need of improvements (30)” with references 30 being the paper mentioned. 

10. No discussion of ethics is complete without an accompanying discussion of the legal framework. Based on my understanding of the Danish law, under the Danish penal code, "all persons must provide aid to the best of their ability to any person who appears to be lifeless or in mortal danger, must alert authorities or take similar steps to prevent impending disasters that could cause loss of life, must comply with all reasonable requests of assistance by a public authority when a person's life, health or well-being is at stake, and must, if they learn of a planned crime against the state, human life or well-being, or significant public goods, do everything in their power to prevent or mitigate the crime, including but not limited to reporting it to authorities, in all cases provided that acting would not incur particular danger or personal sacrifice." Would this not influence the decision-making of first-responders given the law compels them to act?

Author’s response: Thank you for bringing awareness to this area. Future studies should indeed focus on this important aspect of prehospital decision-making. However, in our study, the prehospital physicians were not obligated by law to initiate or continue resuscitation. Please see considerations given above (Reviewer 2, Question no. 1)

---

## [Editor Report · Decision Letter 1]

10 Apr 2023

Ethical considerations in the prehospital treatment of out-of-hospital cardiac arrest: a multi-centre, qualitative study

PONE-D-22-34819R1

Dear Dr. Milling,

We’re pleased to inform you that your manuscript has been judged scientifically suitable for publication and will be formally accepted for publication once it meets all outstanding technical requirements.

Kind regards,

Federica Canzan

Academic Editor

PLOS ONE
---

## [Editor Report · Acceptance letter]

12 Apr 2023

PONE-D-22-34819R1 

Ethical considerations in the prehospital treatment of out-of-hospital cardiac arrest: a multi-centre, qualitative study 

Dear Dr. Milling:

I'm pleased to inform you that your manuscript has been deemed suitable for publication in PLOS ONE. Congratulations! Your manuscript is now with our production department. 

Kind regards, 

on behalf of

Professor Federica Canzan 

Academic Editor

PLOS ONE